# Role of Baseline Gut Microbiota on Response to Fiber Intervention in Individuals with Irritable Bowel Syndrome

**DOI:** 10.3390/nu15224786

**Published:** 2023-11-15

**Authors:** Jerry Zhou, Vincent Ho

**Affiliations:** School of Medicine, Western Sydney University, Campbelltown, NSW 2560, Australia; v.ho@westernsydney.edu.au

**Keywords:** gut microbiome, irritable bowel syndrome, dietary fiber

## Abstract

Irritable bowel syndrome (IBS) is one of the most prevalent functional gut disorders in the world. Partially hydrolyzed guar gum, a low-viscosity soluble fiber, has shown promise in the management of IBS-related symptoms. In this study, we aimed to determine if an individual’s baseline gut microbiota impacted their response to a partially hydrolyzed guar gum intervention. Patients diagnosed with IBS undertook a 90-day intervention and follow-up. IBS symptom severity, tolerability, quality-of-life, and fecal microbiome composition were recorded during this study. Patients with normal microbiota diversity (Shannon index ≥ 3) showed significant improvements to IBS symptom scores, quality-of-life, and better tolerated the intervention compared to patients with low microbiota diversity (Shannon index < 3). Our findings suggest that an individual’s baseline microbiome composition exerts a substantial influence on their response to fiber intervention. Future investigations should explore a symbiotic approach to the treatment of IBS.

## 1. Introduction

The global prevalence of irritable bowel syndrome (IBS) is notably high, estimated to affect approximately 5% to 13% of the general population [1,2]. IBS is classified as a functional disorder characterized by chronic abdominal pain in conjunction with altered bowel habits, which may include urgent diarrhea, chronic constipation, or a pattern of alternating between the two. Presently, the Rome IV Diagnostic Criteria, offering symptom-based criteria, are widely acknowledged as the primary method for diagnosing IBS [3]. The pathogenesis of IBS is intricate and encompasses factors such as intestinal dysmotility, visceral hypersensitivity, and intestinal inflammation. During intestinal inflammation, the intestinal mucus layer and gut barrier functions are impaired, thereby contributing to a constant inflammatory state. Diet and gut microbes have a mutual relationship with the host’s intestinal mucus layer [4,5].

More recently, there has been a growing recognition of the significant role played by the gut microbiome and its fermentation by-products, particularly short-chain fatty acids (SCFA), in the development of IBS. Extensive evidence indicates a reduction in microbiota diversity in individuals with IBS [6], as well as a decrease in fecal SCFA levels compared to samples from healthy subjects [7].

Short-chain fatty acids (SCFAs) are primarily produced in the large intestine through the fermentation of soluble fibers by the gut microbiota and are closely associated with overall gut health and numerous other physiological benefits [8]. One specific type of soluble fiber, partially hydrolyzed guar gum (PHGG, Figure 1), has garnered Grade A recommendations from a consensus group focused on dietary fiber, as it possesses “Level 1 evidence” (based on randomized trials and meta-analyses with a low risk of error) of its clinical efficacy in addressing constipation, diarrhea, and IBS [9]. Notably, PHGG exhibits significantly lower viscosity and is non-fermenting compared to its non-hydrolyzed counterpart while still retaining the beneficial properties of guar gum [9].

Recent research conducted by our group has demonstrated that PHGG maintains its low-viscosity characteristics even under simulated digestion conditions [10]. In a subsequent single-blinded randomized clinical trial, we further established that PHGG provides similar short-term benefits as psyllium husk for blood glucose regulation but is notably more tolerable for individuals with a gut transit disorder [11]. The prebiotic properties of PHGG have been extensively investigated in various in vitro (stool culture) and animal studies [12,13,14,15,16], revealing its ability to create a favorable environment for the growth of beneficial probiotics such as *Lactobacillus* and *Bifidobacterium*. Moreover, PHGG exhibits a unique capacity to undergo extended fermentation within the gut, leading to higher production of SCFAs compared to other dietary fibers [13,17].

Microbial factors play key roles in IBS pathophysiology. A review of IBS studies have shown lower microbial diversity or richness in some, but not all, IBS patients [18]. The relationship between an IBS patient’s baseline microbiota composition and their response to intervention remains unexplored. In this study, we aimed to determine whether the initial gut microbiome of patients with IBS influences their response to PHGG intervention. Over a 90-day intervention period, we closely examined changes in IBS symptoms among patients who consumed 10 g/day of PHGG. In addition, we included a follow-up period of 30 days to evaluate patient responses post-intervention. Our secondary objectives were to assess tolerability, adverse symptoms, and quality-of-life throughout the course of this study.

## 2. Materials and Methods

### 2.1. Recruitment

This study was registered under Australian New Zealand Clinical Trials Registry-Trial ID: ACTRN12621001646831.

Patients aged 18–80 years old with IBS (based on Rome IV criteria) were recruited through word-of-mouth and/or a recruitment poster for this study. Patients were screened through a phone interview, and those that met the eligibility criteria were recruited to take part in the current study. Inclusion criteria were symptomatic IBS within the last 2 weeks, absence of organic gastrointestinal disorder (e.g., Crohn’s disease or ulcerative colitis), and at least one episode of constipation or diarrhea in the last week. Exclusion criteria included usage of the following medications within the last 6 months: opioids or tricyclic antidepressants, antibiotics, probiotics, impaired cognizance and/or legal blindness, pregnant women or nursing mothers. The study was approved by the Western Sydney University Human Research Ethics Committee: H14880. All patients provided written informed consent. The study was carried out at the Macarthur Clinical School, Western Sydney University, Campbelltown, NSW, Australia.

### 2.2. Study Design and Intervention

Participants had a 2 week lead-in period before starting the intervention for 90 days. Following the intervention, patients stopped taking PHGG and were monitored over a 30-day follow-up period. The interventional PHGG (FiberChoice, Nestle Vevey, Vaud, Switzerland) was provided in pre-packed 5 g sealed packets. Patients were asked to consume 10 g/day (2 packets) mixed with 300 mL of water and ingested in the morning at least 30 min before breakfast. Sample size was estimated to be 40 patients in total for this prospective study. The assumed standard deviation of a 3-month IBS Symptom Severity Score (IBS-SSS) score and the smallest relevant clinical difference between groups was IBS-SSS ≥ 50 [19]. The standard deviation for IBS-SSS was based on previous studies [20,21,22,23], and α and β were set to 5% and 20%, respectively.

### 2.3. Questionnaires

The patient-reported symptom score was the IBS-SSS [19], a five-item instrument used to measure the severity of abdominal pain, frequency of abdominal pain, severity of abdominal distension, dissatisfaction with bowel habits, and interference with quality-of-life, each on a 100-point scale. For four of the items, the scales are represented as continuous lines with end points 0 and 100%, with different descriptors at the end points and adverb qualifiers (e.g., “not very”, “quite”) strategically placed along the line. Respondents mark a point on the line between the two end points reflecting the extremity of their judgment. The proportional distance from zero was the score assigned for that scale (hence, scores range from 0 to 100). The end points for the severity items were “no pain” and “very severe”, for satisfaction, the end points were “very satisfied” and “not at all satisfied”, and for interference, they were “not at all interferes” to “completely interferes”. A final item asked the number of days out of 7 the patient experienced abdominal pain, and the answer was multiplied by 10 to create a 0 to 100 metric. The items were summed and thus the total score ranged from 0 to 500. Mild, moderate, and severe cases were indicated by scores of 75 to 175, 175 to 300, and >300, respectively. Patient IBS-SSS scores below 75 were considered to be indicative of remission. Subjective evaluation regarding bowel frequency, stool consistency, and secondary bowel symptoms (pass mucus, presence of blood, hurry/rush, straining, incomplete emptying) were also part of the questionnaire. Patients were asked to complete a weekly lead-in IBS-SSS for the fortnight before commencing the study and completed an IBS-SSS weekly during each study arm, and weekly during the follow-up period. Patients whose total IBS-SSS scores were decreased by ≥50 points after intervention were considered responders.

A tolerability questionnaire scored on consumption (taste, swallowability, overall consumption) and severity of post-ingestion GI symptoms (fullness, pain, nausea, vomiting, flatulence, bloating, burping, sleep disturbance, and general discomfort) was completed by patients at week 2 of the intervention. Each parameter was graded on a 5-point scale: for consumption, the options were “Excellent, Good, Average, Poor, or Bad”, and for the presence of post-ingestion symptoms, the options were “None, Mild, Moderate, Severe, or Very Severe”.

Health-related quality of life was assessed with the English version of the Medical Outcomes Study 36-Item Short Form (SF-36) [24], a measure widely used in clinical practice and research as well as general population surveys. It consists of eight scales that correspond to the main domains of functional status and well-being, including health limitations of physical activities (Physical Functioning), physical health limitations on work and other daily responsibilities (Role Functioning—Physical), intensity of bodily pain or discomfort (Bodily Pain), subjective perception of health status (General Health), physical energy and fatigue (Vitality), impact of health or emotional problems on social activities (Social Functioning), mental health limitations on work and other daily responsibilities (Role Functioning—Emotional), and subjective psychological well-being (Mental Health). Each scale was scored from 0 (poor health) to 100 (optimal health). Scores for the subscales were transformed by dividing actual scores by the maximum score possible and expressing this ratio as a percentage.

### 2.4. Stool Microbiome Profiling

Stool samples were collected using a commercial collection kit, OMNIgene Gut (dnagenotek, Stittsville, ON, Canada) per the manufacturer’s instructions at pre-intervention commencement (day 0), intervention days 30 and 90, and at follow-up (day 120). Samples were mailed to the Australian Genome Research Facility (Adelaide, SA, Australia) for DNA isolation and microbiome sequencing. Stool samples were homogenized and DNA extracted using the QIAamp Stool Mini Kit (Qiagen, Hilden, Germany). Amplicon sequencing was performed targeting the hypervariable region (V1–V3, 27F/529R) of the 16S rRNA gene on the Illumina MiSeq platform (Illumina, San Diego, CA, USA), using the Illumina Nextera XT Index with paired-end sequencing. Raw paired-end Illumina reads were trimmed using Cutadapt. Sequence analysis was performed using Quantitative Insights into Microbial Ecology 2 (QIIME Version 8.0.1623). QIIME was also used to generate amplicon sequence variants (ASVs). Sequences were normalized to relative abundance of reads per million. The final total number of ASVs following quality filtering was 3145 reads. Sequences were then clustered into operational taxonomic units (OTUs) following the default QIIME2 pipeline based on 99% sequence similarity against the Greengenes database, version 13.8. Alpha diversity metrics were determined by the Shannon index. Beta diversity was analyzed based on Bray–Curtis and Jaccard distances. Permutational multivariate analysis of variance (PERMANOVA), linear discriminant analysis (LDA), and effect size (LEfSe) analysis were performed in PRIMER version 6 (PRIMER-E, UK). Visualization was performed using non-metric multidimensional scaling (nMDS). Comparisons of relative abundance at the taxonomy levels across different groups were performed in the R package “mvabund” using multivariate generalized linear models (GLM) assuming a negative binomial distribution. Significant pairs were identified using the pairwise LEfSe. Patients and investigators were blinded to the microbiome diversity data during the study. Data were unblinded at the study’s conclusion for analysis. Normal microbiota was defined as patients having a baseline (day 0) gut microbiota alpha diversity of ≥3.0 (Shannon index), and low microbiota was defined as patients having a baseline (day 0) gut microbiota alpha diversity of <3.0 (Shannon index) [25,26].

### 2.5. Statistical Analysis

Data are presented as mean ± standard deviation, unless otherwise specified. Symptoms and quality-of-life values were assessed for normality distribution before either parametric two-way ANOVA or non-parametric Wilcoxon’s paired sample test was conducted. Instat (version 3.1) for Windows statistics software package (Graphpad Software, San Diego, CA, USA) was used. *p* ≤ 0.05 was considered statistically significant.

## 3. Results

### 3.1. Recruitment

A total of 49 symptomatic IBS patients (any IBS sub-type) were recruited and screened. The first subject in was on 12 July 2022 and last subject out was on 4 August 2023. Of these, 40 (82%) completed the study, 1 (2%) was excluded due to being in remission (IBS-SSS < 75) during the lead-in period, 2 (4%) were lost to follow-up, 2 (4%) were excluded as they had taken a prohibited medication (antibiotics) during the study period, 2 (4%) were unable to continue in the study due to unforeseen travel commitments, and 2 (4%) were excluded due to incomplete questionnaires.

Patients recruited were predominantly female (76%) with an average age of 48.7 years old and average BMI of 28.0. Patient dietary screening indicated an average of 2.9 daily servings of fruit and vegetables, lower than the recommended 4–5 servings. All participants were diagnosed with IBS, and all but one were symptomatic during the two weeks before the study. The majority of participants had moderate to severe IBS-SSS (71%) at baseline. Thirty-two patients (61%) had low microbiome diversity, while nineteen (39%) had normal microbiome diversity. The characteristics of patients who completed the study are summarized in Table 1.

The microbiota diversity increased in response to intervention within the low baseline diversity group. During the follow up period, there was a statistically significant increase in the Shannon Index compared to baseline (*p* = 0.0164). A slight reduction in the Shannon Index was noted in the normal diversity group but this change was not statistically significant.

### 3.2. IBS Symptoms Severity Scores

The changes in the IBS-SSS and PHGG response rate during intervention and follow-up are presented in Table 2. Baseline IBS-SSS for both microbiome groups were within the “moderate” range (Moderate IBS-SSS 175–300). The normal diversity group showed significant reductions in IBS-SSS compared to baseline at day 30 (35%), day 90 (47%), and day 120 (36%). During intervention and follow-up, IBS-SSS in the normal diversity group was reduced to within the “mild” range (Mild IBS-SSS 75–174). The low diversity group did not show significant reductions compared with baseline. The normal diversity group also had a higher number of patients responding to the PHGG intervention (reduction of 50 or more IBS-SSS) compared with those in the low diversity group at intervention and follow-up.

A moderate, positive correlation (R^2^ = 0.493) was noted within the normal diversity group between a participant’s Shannon Index and percentage IBS symptom response (Appendix A). This trend was not observed in the low diversity group, where there was a poor correlation between Shannon Index and IBS symptom response (Appendix A).

### 3.3. Differential Microbiome Composition

The groups had significantly different Shannon diversity indexes at baseline (Table 1) but this difference was no longer evident at days 30, 90, and follow-up day 120. Microbiota diversity in both the normal and low diversity groups did not significantly change during the study.

LEfSe analysis identified the abundance of specific microbial taxa between diversity groups (Table 3). At the phylum level, Actinobacteria was significantly more abundant in the normal diversity group than in the low diversity group. At the genus level, Oscillospira and Odoribacter were significantly more abundant in the normal diversity group than in the low diversity group. At the species level, *F. pausnitzii* and *P. copri* were significantly less abundant in the normal diversity group compared to the low diversity group.

Participants were also grouped into responsive (reduction of >50 IBS-SSS) and non-responsive (reduction of ≤50) to PHGG. The PHGG responsive group had a mean percentage symptom reduction of 38% (SD ± 12.6) and baseline Shannon Index of 2.87 (SD ± 0.48). Meanwhile, the PHGG non-responsive group had a mean percentage symptom reduction of 1% (SD ± 24.7) and baseline Shannon Index of 2.32 (SD ± 0.70). Within the PHGG responsive group, 86% also had a normal baseline microbiota diversity (Shannon Index ≥ 3), while in the non-responsive group only 34% had a normal baseline microbiota diversity. The differential microbes between these groups are presented in Appendix A. In the PHGG responsive group, composition of *Oscillospira* and *Odoribacter* were significantly higher than in the non-responsive group. Conversely, *F. prausnitzii*, *P. copri*, and *Prevotella* were significantly reduced in the PHGG responsive group compared with the non-responsive group. These differential microbes are similar to those identified between the baseline diversity groups.

### 3.4. Tolerability

The tolerability of PHGG at a dosage of 10 g/day is presented in Figure 2. In the normal diversity group, patients rated PHGG taste, swallowability, and overall consumption as excellent, good or average at 100%, 90%, and 100%, respectively. In the low diversity group, patients rated PHGG taste, swallowability, and overall consumption slightly lower at 90%, 90%, and 85%, respectively.

Gastrointestinal symptoms experienced after PHGG ingestion are presented in Figure 3. The most common symptoms reported as “severe” or “very severe” were flatulence (25% normal diversity and 20% low diversity) and bloating (25% normal diversity and 15% low diversity). In the low diversity group, fullness, sleep disturbance, and general discomfort were also noted; these symptoms were all reported by 5% of patients as “severe” or “very severe”.

### 3.5. Quality of Life

Quality-of-life (SF-36) during the study from the normal and low microbiota diversity groups are shown in Table 4. There was no significant difference between groups based on their baseline quality-of-life subscale scores.

The normal diversity group showed significant improvement in the subscales Role-Emotional, Mental Health, Social Functioning and General Health at day 30 compared with baseline. In addition to these improvements at day 30, day 90 saw significant improvements in Physical Function, Role–Physical, and Bodily Pain. During follow-up, all subscale improvements were sustained except for Social Functioning and Bodily Pain. In the low diversity group, only Role–Physical showed significant improvement (at day 90 of intervention).

## 4. Discussion

The findings from our current study underscore the importance of an IBS patient’s baseline gut microbiota in response to PHGG intervention. Notably, individuals with a baseline microbiota diversity falling within the normal range (Shannon index ≥ 3) experienced the most significant and sustained symptomatic and quality-of-life improvements. Conversely, participants with a lower baseline diversity exhibited comparatively fewer benefits from PHGG intervention

IBS stands out as one of the most prevalent diagnoses within functional gastrointestinal disorders. All patients included in our study met the Rome IV criteria for IBS; in addition, thorough blood tests and endoscopic examinations were conducted to rule out any organic causes for their symptoms. The effectiveness of PHGG in alleviating IBS-related symptoms is well-established [27,28]. In a previous investigation [29], PHGG was administered at two different dosages, 5 and 10 g/day, to patients with IBS. The results indicate that PHGG treatment led to improvements in symptoms and quality-of-life during the first month of treatment. These benefits declined by the 3-month intervention mark and in the subsequent 6-month follow-up, albeit remaining for the most part significantly above baseline.

Intestinal dysbiosis has been frequently linked to patients with IBS. While some studies have reported lower microbial diversity in IBS patients compared to healthy controls, this trend is not consistently observed in all studies [18]. In our current study, we observed low gut microbiome diversity in 65% of our IBS patients. At the phylum level, there was a significantly lower proportion of Actinobacteria in the low diversity group than in the normal diversity group. At the genus level, the normal diversity group exhibited significantly lower proportions of *Prevotella* and *Faecalibacterium* while composition of *Oscillospira* and *Odoribacter* were higher compared to the low diversity group. At the species level, *P. copri* and *F. prausnitzii* were lower in the normal diversity group compared to these species in the low diversity group. These findings suggest that an individual’s response to PHGG may be associated with their gut microbiota’s diversity and composition.

Notably, *F. prausnitzii* and *P. copri* comprised 33.2% of the microbiome in the low diversity patients, which was 4.6-fold higher than in the normal diversity patients (7.2%). *Prevotella* species, anaerobic Gram-negative bacteria in the Bacteroidetes phylum, have been linked to *Prevotella*-enriched gut dysbiosis associated with chronic inflammatory diseases, often leading to inflammation mediated by T helper 17 cell-related immune responses [30]. Some strains of *Prevotella* have even been suggested to be clinically significant pathobionts that can exacerbate chronic inflammation. In contrast, *F. prausnitzii*, an anaerobic Gram-positive bacterium, is considered an anti-inflammatory commensal bacterium [31]. A low count of *F. prausnitzii* is typically found in patients with colitis, and the anti-inflammatory metabolites produced by this bacterium have been detected in patients with Crohn’s disease [32,33]. Interestingly, a similar increase in the abundance of these two bacterial species has been observed in patients with functional abdominal bloating and distension compared to healthy controls [34]. The reasons for the paradoxical results among IBS patients concerning the abundance of *P. copri* and *F. prausnitzii*, which play opposing roles in the gut microbiome, remain unexplored.

An increase in genera, *Oscillospira* and *Odoribacter*, observed in the normal diversity group may play crucial roles in the utilization of PHGG in the gut. *Oscillospira*, which is currently only described in high-throughput sequencing data related to the human gut microbiota, has yet to be cultured, and its biological function and specific role in human health remain elusive. *Oscillospira* is generally a slow-growing organism and has been associated with slow transit constipation, a condition in which slower-replicating organisms persist in the gut and avoid being eliminated [35]. Metabolic profiling suggests that *Oscillospira* is likely capable of producing SCFA, which may contribute to its role in reducing inflammation in the gut. Furthermore, there is evidence indicating that *Oscillospira* may have lipid-lowering and metabolic syndrome-alleviating effects, as it exhibits positive regulatory effects in obesity and metabolic diseases [36].

*Odoribacter*, on the other hand, has been shown to be crucial for maintaining a healthy gut, and decreased abundance is associated with conditions such as inflammatory bowel disease, non-alcoholic fatty liver disease, and cystic fibrosis [37,38,39]. Specific species of *Odoribacter*, such as *O. splanchnicus* and *O. laneus*, have been found to improve glucose tolerance and reduce inflammation by producing outer membrane vesicles and reducing intestinal succinate [40,41]. Nevertheless, the cause-and-effect relationship between microbiota composition and response to PHGG in IBS remains a topic that requires further investigation.

IBS has a well-documented adverse impact on health-related quality of life, with the alleviation of psychosocial dysfunction and symptom relief ranking among the primary objectives of IBS treatment [42]. Numerous studies employing the SF-36 assessment tool have consistently demonstrated that individuals seeking treatment for IBS exhibit significantly lower quality-of-life scores than the general population [43,44,45]. In alignment with these findings, our IBS patients initially displayed compromised quality-of-life, as evidenced by SF-36 scale scores similar to those reported in previous studies. Notably, our study documented significant improvements in quality-of-life scores in the normal microbiota diversity group, which were largely sustained after intervention into the follow-up period. A normal diversity microbiota also confers better tolerability and less severe gastrointestinal symptoms during PHGG intervention. It has been shown that a diverse and rich gut microbiota enables the appropriate break down of fibers into readily absorbed SCFAs, while dysbiosis can prolong fermentation promoting the production of excess hydrogen and carbon dioxide [46].

Several limitations should be considered when interpreting the results of this study. First, it is important to acknowledge the potential influence of dietary factors on gut microbiota composition and changes in IBS symptoms. While patients were screened for their dietary habits and instructed to maintain their usual dietary patterns during the trial, this study did not impose strict dietary controls. Although the study groups exhibited similar dietary patterns at the outset, the lack of stringent dietary control during the study could introduce variability. Secondly, the normal diversity group did not contain any patients with the diarrhea predominate IBS subtype, which made up 27% of the low diversity group. Given the size of our normal diversity group, we should expect to have three to four patients with the diarrhea predominant phenotype. This may affect the applicability of our findings to this IBS subtype. A recent systematic review of studies of gut microbiota in IBS patients found no consistent differences in diversity or bacteria taxa between IBS subtypes in the fourteen studies that were evaluated [47]. Therefore, the absence of IBS-diarrhea patients may not impact the microbiota composition of the group. Thirdly, it is worth noting that our study was not placebo-controlled. Given the high placebo response rates commonly observed in IBS trials [48], it is possible that patient expectations of improvement due to the introduction of a new product could have influenced our results. However, it is important to emphasize that both the investigators and the patients were blinded to the microbiome composition, and the diversity groups exhibited significantly different responses. Therefore, the absence of a placebo group may not have introduced bias or significantly influenced this study’s outcomes.

## 5. Conclusions

In summary, our findings suggest that an individual’s baseline microbiome composition exerts a substantial influence on their response to PHGG for IBS intervention. Patients with a normal baseline microbiome diversity not only experienced a significant reduction in IBS symptoms and improvements to their quality-of-life but also exhibited better tolerance to PHGG compared to their low diversity counterparts. Future investigations may explore a symbiotic approach involving the combination of specific or multiple probiotics with PHGG to determine whether this approach offers additional benefits to individuals with IBS.

## Figures and Tables

**Figure 1 nutrients-15-04786-f001:**
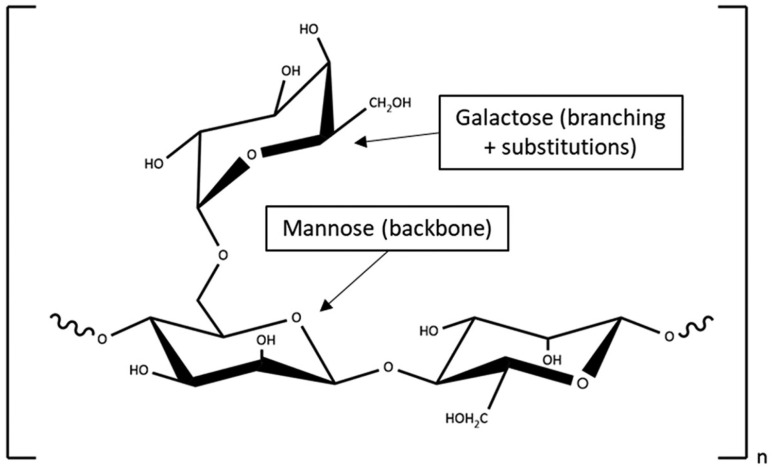
Structure of the main partially hydrolyzed guar gum (PHGG) polysaccharide (galactomannan). Guar gum is enzymatically hydrolyzed to produce PHGG, consisting of short (3–8 monomers) and medium (9–30 monomers) polysaccharide chains at a 1:7 ratio.

**Figure 2 nutrients-15-04786-f002:**
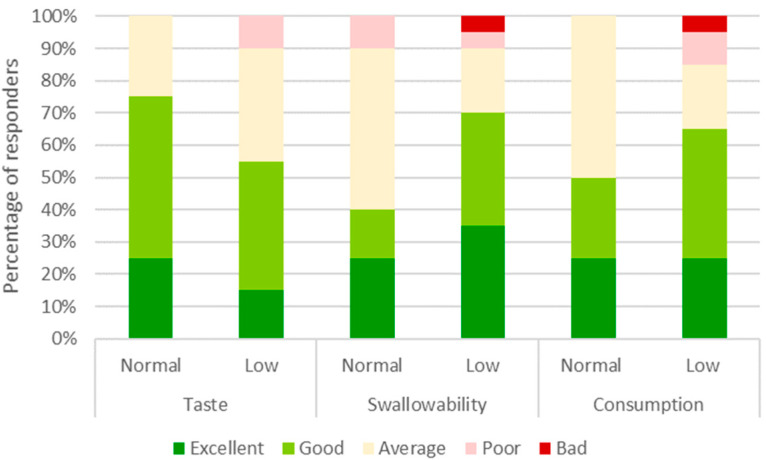
Tolerability of partially hydrolyzed guar gum for patients with normal microbiota diversity (Normal, *n* = 14) and low microbiota diversity (Low, *n* = 26). Participants evaluated taste, swallowability, and overall difficulty of consumption on a 5-point scale (Excellent, Good, Average, Poor, and Bad). The number of responses in each category were converted to a percentage of the cohort.

**Figure 3 nutrients-15-04786-f003:**
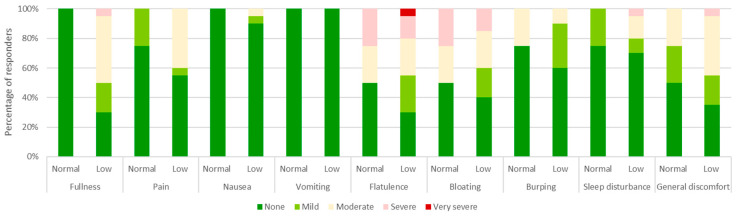
Severity of gastrointestinal symptoms experienced after ingestion of partially hydrolyzed guar gum in patients with normal microbiota diversity (Normal, *n* = 14) and low microbiota diversity (Low, *n* = 26). The number of responses in each category were converted to a percentage of the cohort.

**Table 1 nutrients-15-04786-t001:** Characteristics of patients who completed the study.

Characteristics	Normal Diversity (*n* = 14)	Low Diversity (*n* = 26)	*p* Value ^#^
Female gender *n* (%)	12 (79%)	20 (77%)	
Age, mean years ± SD	52.0 ± 17.8	49.0 ± 16.5	0.677
BMI, mean ± SD	32.0 ± 6.7	27.4 ± 6.1	0.102
Diagnosed IBS subtype *n* (%)			
Constipation Predominant	10 (71%)	6 (23%)	
Diarrhea Predominant	0	7 (27%)	
Alternating/Mixed	4 (29%)	13 (50%)	
Baseline daily servings of fruits and vegetables, mean ± SD	2.9 ± 1.8	3.0 ± 1.8	0.859
Baseline microbiota diversity (Shannon index ± SD)			
Baseline (Day 0)	3.2 ± 0.1	2.3 ± 0.6	>0.001
Intervention (Days 1–30)	2.8 ± 0.7	2.5 ± 0.4	0.353
Intervention (Day 31–90)	2.8 ± 0.4	2.5 ± 0.5	0.295
Follow-up (Days 91–120)	2.8 ± 0.4	2.8 ± 0.4 *	0.984

^#^ *p*-value comparison between normal diversity and low diversity group characteristics; * Significant (<0.05) difference compared baseline (Day 0).

**Table 2 nutrients-15-04786-t002:** IBS symptom severity scores during study time points and response rate at intervention and follow-up in patient groups with normal and low gut microbiome diversity. Comparisons between diversity groups are denoted.

	Normal Diversity (*n* = 14)	Low Diversity (*n* = 26)	*p* Value ^#^
IBS-SSS			
Baseline (Day 0) ± SD	240 ± 44	258 ± 98	0.080
Intervention (Days 1–30) ± SD	155 ± 57 **	219 ± 85	0.042
Intervention (Day 31–90) ± SD	128 ± 56 **	194 ± 108	0.064
Follow-up (Days 91–120) ± SD	153 ± 79 *	227 ± 123	0.127
Response rate ^			
Intervention (Days 1–30)	10 (71%)	9 (35%)	
Intervention (Day 31–90)	12 (86%)	15 (57%)	
Follow-up (Days 91–120)	10 (71%)	11(42%)	

Partially hydrolyzed guar gum (PHGG), Irritable bowel syndrome symptom severity score (IBS-SSS); ^#^
*p*-value comparison between normal diversity and low diversity group; ^ Patient response to intervention is defined as a reduction of at least 50 IBS-SSS from baseline; * Significant (<0.05) difference compared to its respective Baseline (Day 0); ** Significant (<0.01) difference compared to its respective Baseline (Day 0).

**Table 3 nutrients-15-04786-t003:** Differential abundance of microbial taxa between baseline microbiota diversity groups.

Phylum	Bacteria	Normal Diversity (% ± SD)	Low Diversity (% ± SD)	*p* Value *	FDR	LDA Score
Actinobacteria		1.43 ± 2.32	0.18 ± 0.23	<0.001	<0.001	4.82
Firmicutes	*Oscillospira*	5.07 ± 4.56	1.23 ± 0.81	<0.001	0.002	5.28
Bacteroidetes	*Odoribacter*	2.05 ± 3.59	0.16 ± 0.13	<0.001	0.007	4.98
Firmicutes	*F. prausnitzii*	7.17 ± 5.98	19.46 ± 15.14	0.0033	0.0357	−5.79
Bacteroidetes	*P. copri*	0.07 ± 0.13	13.80 ± 28.13	0.0034	0.0422	−5.84
Firmicutes	*Faecalibacterium*	8.78 ± 6.17	20.12 ± 15.82	0.0182	0.0153	−5.11
Bacteroidetes	*Prevotella*	0.09 ± 0.19	15.81 ± 27.61	0.0035	0.0421	−5.90

* Differential analysis using LEfSe (*p* value < 0.05 and false discovery rate < 0.05).

**Table 4 nutrients-15-04786-t004:** Quality-of-life (SF-36) scores for patients over the course of the trial.

Normal Diversity Group (*n* = 14)				
**Subscale**	**Day 0 ± SD**	**Day 30 ± SD**	**Day 90 ± SD**	**Day 120 ± SD**
Physical Function	72.1 ± 21.1	83.2 ± 21.5	92.8 ± 19.7 *	90.1 ± 21.4 *
Role–Physical	68.7 ± 22.6	72.0 ± 26.9	87.2 ± 27.2 *	82.8 ± 23.3 *
Role–Emotional	71.6 ± 22.1	82.8 ± 25.8 *	86.3 ± 21.3 *	88.2 ± 14.3 *
Vitality	45.8 ± 21.8	51.5 ± 23.1	62.6 ± 21.1	61.9 ± 23.1
Mental Health	74.1 ± 14.8	85.8 ± 17.9 *	87.3 ± 18.4 *	81.6 ± 13.4 *
Social Functioning	71.7 ± 23.7	88.1 ± 18.2 *	92.2 ± 17.6 *	84.7 ± 25.1
Bodily Pain	64.2 ± 16.5	76.5 ± 27.9	79.8 ± 22.2 *	76.7 ± 23.6
General Health	52.1 ± 21.8	67.3 ± 19.9 *	74.4 ± 21.2 *	73.1 ± 22.0 *
**Low Diversity Group** (***n* = 26**)				
**Subscale**	**Day 0 ± SD**	**Day 30 ± SD**	**Day 90 ± SD**	**Day 120 ± SD**
Physical Function	73.3 ± 23.7	77.7 ± 25.4	75.3 ± 29.1	76.6 ± 26.3
Role–Physical	66.4 ± 21.8	62.5 ± 21.5	76.8 ± 20.1 *	62.5 ± 22.9
Role–Emotional	72.3 ± 17.3	57.1 ± 10.9	67.6 ± 19.0	63.3 ± 18.3
Vitality	47.9 ± 22.3	46.1 ± 25.4	48.7 ± 22.1	42.7 ± 28.1
Mental Health	75.5 ± 28.5	74.8 ± 22.5	77.2 ± 27.1	75.0 ± 26.9
Social Functioning	69.0 ± 29.0	71.8 ± 28.1	78.8 ± 24.1	72.7 ± 25.9
Bodily Pain	61.2 ± 17.3	61.4 ± 20.7	67.4 ± 21.2	59.1 ± 22.3
General Health	55.8 ± 19.8	57.1 ± 22.9	53.0 ± 18.3	57.5 ± 23.5

* Difference vs. baseline significant at *p* < 0.05.

## Data Availability

Data files underpinning the research presented in this article are available from the corresponding author upon request.

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
