# Peer review of "Role of Baseline Gut Microbiota on Response to Fiber Intervention in Individuals with Irritable Bowel Syndrome"

_nutrients, 2023, doi:10.3390/nu15224786_

Round 1
Reviewer 1 Report
Comments and Suggestions for Authors
The authors aimed to investigate gut microbiota in IBS patients in association with partially hydrolyzed guar gum (PHGG) intervention. Patients with normal basal microbiota diversity showed better response to PHGG intervention compared to patients with low microbiota diversity. It contains some interesting observations but is totally descriptive without any functional assays. In addition, the sample size was small, and neither placebo control nor healthy control was analyzed. So, it is hard to draw a conclusion and I'm concerned if the obtained results are reproducible. My comments are shown below.
l Please describe the definition of “normal” diversity. How and who defines “Shannon index≥3” as a “normal” diversity? Was it determined by the data from healthy individuals in another study? (Almost all healthy individuals have “normal” microbial diversity?)
l Likewise, are there any patients with high microbial diversity who had a good response to PHGG in the “normal” diversity? Please show a correlation chart in terms of levels of microbial diversity and IBS-SSS.
l It is interesting to compare the microbiota in IBS patients with or without good response to PHGG intervention.
l Why is the low baseline microbiota diversity tolerable to PHGG intervention? I guess some PHGG-related bacteria are missing in those patients.
l Is the low baseline microbiota diversity pathogenic in IBS?
l Some metabolome assays should be considered to reveal the mechanisms associated with successful intervention of PHGG.
Author Response
The authors would like to thank the reviewer for their comments and feedback. We have provided our responses to the comments below:
1. Please describe the definition of “normal” diversity. How and who defines “Shannon index≥3” as a “normal” diversity? Was it determined by the data from healthy individuals in another study? (Almost all healthy individuals have “normal” microbial diversity?)
The cut-off value for diversity was based on thresholds from previous clinical (Yin et al. 2019) and population studies (Li et al. 2012). The manuscript has been updated to include these references - Methods [Line 161].
Yin, L., Wan, Y.D., Pan, X.T., Zhou, C.Y., Lin, N., Ma, C.T., Yao, J., Su, Z., Wan, C., Yu, Y.W. and Zhu, R.X., 2019. Association between gut bacterial diversity and mortality in septic shock patients: a cohort study. Medical Science Monitor: International Medical Journal of Experimental and Clinical Research, 25, p.7376.
Li, K., Bihan, M., Yooseph, S. and Methe, B.A., 2012. Analyses of the microbial diversity across the human microbiome. PloS one, 7(6), p.e32118.
2. Likewise, are there any patients with high microbial diversity who had a good response to PHGG in the “normal” diversity? Please show a correlation chart in terms of levels of microbial diversity and IBS-SSS.
The reviewer raises an interesting question and the authors have conducted further analysis to evaluate correlation between diversity and PHGG response in the two groups. We found that within the normal baseline diversity cohort a moderate positive correlation was noted (R2 = 0.493) between diversity score and reductions in symptom severity (Fig. S1). Conversely, in the low diversity cohort, no correlation was detected between diversity and symptom severity (R2 = 0.0026; Fig. S2). This analysis was added to the manuscript – Results [Line 217 – 221] and Supplementary Materials as Figures S1 and S2.
3. It is interesting to compare the microbiota in IBS patients with or without good response to PHGG intervention.
This is a good query raised by the reviewer and the authors have conducted further analysis establish PHGG response groups and evaluate microbiota differences. We defined the PHGG responsive group to exhibit a symptom reduction of >50 IBS-SSS on trial Day 30 while the non-responsive group has a symptom reduction of ≤50 IBS-SSS on trial Day 30. Within the PHGG responsive group, 86% also had a normal baseline microbiota diversity (Shannon Index ≥3), while in the non-responsive group only 34% had a normal baseline microbiota diversity.
In the PHGG responsive group, composition of Oscillospira and Odoribacter were significantly higher than in the non-responsive group. Conversely, F. prausnitzii, P. copri, and Prevotella were significantly reduced in the PHGG responsive group compared with the non-responsive group. These differential microbes are similar to those identified between the baseline diversity groups shown in Table 3.
The additional analysis has been updated in the manuscript – Results [Line 238 – 248] and Supplementary Materials Table S1.
4. Why is the low baseline microbiota diversity tolerable to PHGG intervention? I guess some PHGG-related bacteria are missing in those patients.
Although PHGG was less well tolerated by the low diversity group than their normal diversity counterpart, the majority of patients in both groups were able to tolerate the fiber. We hypothesis that in the low diversity patients key PHGG-related bacteria are absent or in low numbers, the fiber would not be fully utilised resulting in partial or no fermentation. This would explain the similar levels of tolerability experienced by the diversity groups.
5. Is the low baseline microbiota diversity pathogenic in IBS?
A low baseline microbiota diversity is likely a symptom of IBS rather than the cause. Emerging evidence suggests a combination of psychological factors (e.g. stress anxiety), infection, and antibiotic usage are risk factors for IBS. Studies comparing the microbiota populations in IBS patients to healthy controls show some, but not all, had lower microbial diversity or richness in IBS patients versus health controls. A recent study, using a machine learning procedure, showed severe IBS was more likely to be associated with decreased microbial diversity. Although altered microbiota diversity is often noted in IBS patients it is unlikely to be the direct cause of IBS.
6. Some metabolome assays should be considered to reveal the mechanisms associated with successful intervention of PHGG.
The mechanism of action for PHGG in symptom relief has been attributed to slow fermentation leading to the production of beneficial SCFAs and prevention of gases and methane. Strong prebiotic affects have also been observed and a reduction in harmful Clostridium species. Given we noted differences in several butyrate-producing bacteria between the groups, it would indeed be beneficial to explore mechanisms further in future studies to determine an association between baseline microbiota diversity with the production of beneficial/detrimental metabolites.
Reviewer 2 Report
Comments and Suggestions for Authors
The role of dietary pathogenesis of IBS is still poorly known. This justifies the need for further research in this area.
The weakness of the study is the heterogenous group of patients, because the pathogenesis of different types of IBS varies.
The value of the research would be greater using a placebo control group.
Moreover, the relationship between dietary factors and gut bacteria requires assessment of the microbiota status before and after nutritional intervention.
I assume that you take my critical comments into account in further research.
Generally, the work has some scientific value and practical usefulness. The manuscript is well prepared in terms of content, editorial, and statistical analysis and can be published in presented form.
Author Response
We like to thank the reviewer for their comments and feedback. There is indeed scope for further investigation into the relationship between diet and IBS.
The heterogeneity of the IBS patients do pose a limitation for the application of our findings. From our results, IBS-D patients have lower microbiota diversity but also showed statistically significant improvements to microbiota diversity post PHGG treatment. This leads to the hypothesis that a future study within IBS-D patients would introduce a longer intervention period, which may be required for their microbiota diversity to be restored and benefits from PHGG.
Regarding the inclusion of a placebo control group, future studies would warrant the inclusion of a control group given a placebo fiber. Theoretically, differences between their baseline microbiota should not have a impact on symptoms changes in these groups.
Reviewer 3 Report
Comments and Suggestions for Authors
In this manuscript, Zhou et al. have investigated in IBS individuals the implication of a baseline microbiota composition to the response of a dietary fiber (i.e., guar gum) intervention. They found out the importance of the gut microbiota in such dietary treatment, but future investigations should address a symbiotic approach to the treatment of IBS. The manuscript is well-written, and I highly appreciate that the authors have outlined the key limitations of this study. However, I found that the introduction lack important information and the authors must slightly extend it to harmonize this part with the discussion part. Please see my comments below:
Major comments:
I will slightly extend the introduction part by explaining that during intestinal inflammation, the intestinal mucus layer and the gut barrier function are impaired, thereby contributing to a constant inflammatory state. There are two recent reviews that have recently revised the link between the diet-microbiota-intestinal mucus layer in the context of intestinal inflammation (Suriano et al., Frontiers Immunology 2022), and the mutual relationship between microbes and the mucus layer (Luis&Hansson, Cell Host Microbe 2023), respectively. Please add these two references in the introduction. This will also help the authors to connect the intro part with the discussion part (lines 312-313) in which authors have underlined the importance of Odoribacter in maintaining a healthy gut. However, it is not clear whether dysbiosis is the initiating factor in IBS (I will underline this important point in the introduction as well).
Minor comments:
The name of the bacteria at the species/genus level should be in Italics.
In vitro and in vivo should go in Italics as well.
Author Response
The authors would like to thank the reviewer for their insight and feedback. Please see our response to reviewer comments below:
Major comments:
I will slightly extend the introduction part by explaining that during intestinal inflammation, the intestinal mucus layer and the gut barrier function are impaired, thereby contributing to a constant inflammatory state. There are two recent reviews that have recently revised the link between the diet-microbiota-intestinal mucus layer in the context of intestinal inflammation (Suriano et al., Frontiers Immunology 2022), and the mutual relationship between microbes and the mucus layer (Luis&Hansson, Cell Host Microbe 2023), respectively. Please add these two references in the introduction. This will also help the authors to connect the intro part with the discussion part (lines 312-313) in which authors have underlined the importance of Odoribacter in maintaining a healthy gut. However, it is not clear whether dysbiosis is the initiating factor in IBS (I will underline this important point in the introduction as well).
Intestinal inflammation has been further described with the addition of the two reference in the Introduction [Line 60 – 64] and role of dysbiosis in IBS.
Minor comments:
The name of the bacteria at the species/genus level should be in Italics.
Bacteria species/genus names have been italics in manuscript.
In vitro and in vivo should go in Italics as well.
These have been updated in the manuscript.
Round 2
Reviewer 1 Report
Comments and Suggestions for Authors
The authors answered most of my critiques and the manuscript has been improved.
Author Response
We would like to thank the reviewer for their feedback. No additional changes were required.
Reviewer 3 Report
Comments and Suggestions for Authors
The authors have addressed all the reviewer´s comments. I have only one minor comment:
Please rephrase lines 33-34. Do you want to say that diet and gut microbes have a mutual relationship with the host´s intestinal mucus layer? If so, please replace for with have.
Author Response
We like to thank the reviewer for the comments. Please find amended manuscript - Introduction Line 33 - 34.